# Lineage tracing of human B cells reveals the in vivo landscape of human antibody class switching

**Felix Horns[1†], Christopher Vollmers[2,3†], Derek Croote[2], Sally F Mackey[4], Gary E Swan[5,6], Cornelia L Dekker[4], Mark M Davis[7,8], Stephen R Quake[2,9,10*]**

[1]Biophysics Graduate Program, Stanford University, Stanford, United States; [2]Department of Bioengineering, Stanford University, Stanford, United States; [3]Department of Biomolecular Engineering, University of California Santa Cruz, Santa Cruz, United States; [4]Department of Pediatrics, Stanford University School of Medicine, Stanford, United States; [5]Stanford Prevention Research Center, Stanford University School of Medicine, Stanford, United States; [6]Department of Medicine, Stanford University School of Medicine, Stanford, United States; [7]Department of Microbiology and Immunology, Stanford University School of Medicine, Stanford, United States; [8]Institute of Immunity, Transplantation and Infection, Stanford University School of Medicine, Stanford, United States; [9]Department of Applied Physics, Stanford University, Stanford, United States; [10]Howard Hughes Medical Institute, Chevy Chase, United States

**\*For correspondence:** quake@ stanford.edu

[†]These authors contributed equally to this work

**Competing interests:** The authors declare that no competing interests exist.

**Abstract** Antibody class switching is a feature of the adaptive immune system which enables diversification of the effector properties of antibodies. Even though class switching is essential for mounting a protective response to pathogens, the in vivo patterns and lineage characteristics of antibody class switching have remained uncharacterized in living humans. Here we comprehensively measured the landscape of antibody class switching in human adult twins using antibody repertoire sequencing. The map identifies how antibodies of every class are created and delineates a two-tiered hierarchy of class switch pathways. Using somatic hypermutations as a molecular clock, we discovered that closely related B cells often switch to the same class, but lose coherence as somatic mutations accumulate. Such correlations between closely related cells exist when purified B cells class switch in vitro, suggesting that class switch recombination is directed toward specific isotypes by a cell-autonomous imprinted state.

## Introduction

The human immune system's antibody repertoire provides broad protection against pathogen infection. The variable regions of antibodies have been the subject of intense study due to their central role in determining the amazing breadth of molecular recognition in the antibody repertoire. However, the constant regions of antibodies also display quite dynamic behavior through the phenomenon of class switching, which is also known as isotype switching. Different classes of antibodies with distinct Fc domains mediate specialized effector functions, including activation of complement, phagocytosis, cytotoxicity, and release of inflammatory mediators (*Kindt et al., 2007*). The diversification of antibody functionality via class switching is essential for mounting a protective response to different pathogens. Conversely, dysregulation of antibody class switching has been implicated in autoimmune diseases, including allergic hypersensitivity (*Sugai et al., 2003*),

**eLife digest** The human immune system comprises cells and processes that protect the body against infection and disease. B cells are immune cells that once activated produce antibodies, or proteins that help identify and neutralize infectious microbes and diseased host cells.

Antibodies fall into one of ten different classes, and each class has a different, specialized role. Certain antibody classes are responsible for eradicating viruses, while others recruit and help activate additional cells of the immune system.

B cells multiply quickly once they are activated. During this proliferation process, dividing B cells can switch from making one class of antibody to another. As such, a single activated B cell can yield a group of related B cells that produce distinct classes of antibodies. Although much has been learned about antibody class switching and its role in generating a diverse set of antibodies, the process of creating different antibody classes in humans remains unknown.

Horns, Vollmers et al. now reveal how antibodies of every class are created in living humans. By developing a way to reconstruct the B cell proliferation process and thereby trace the lineage of individual B cells, the occurrence of class switching events could be measured and mapped. This approach revealed that most antibodies are produced via a single dominant pathway that involves first switching through one of two antibody classes.

Horns, Vollmers et al. also determined that closely related B cells, which were recently born through division of a common ancestor, often switched to the same class. The shared fate is likely explained by the existence of similar conditions inside each cell, which are inherited during cell division and direct switching toward a particular class. All together, these new findings lay a foundation for developing techniques to direct antibody class switching in ways that support the immune system. Future work will aim to understand the conditions inside a cell that direct switching toward a particular class of antibody.

rheumatoid arthritis (*Humby et al., 2009*), systemic lupus erythematosus (*Bubier et al., 2009*; *Mietzner et al., 2008*), IgG4-related disease (*Stone et al., 2012*), and hyperimmunoglobulin E syndrome (*Minegishi, 2009*).

Class switching occurs during germinal center maturation and is linked to cell division and somatic hypermutation (*Hodgkin et al., 1996*; *Liu et al., 1996*; *Tangye et al., 2002*). After antigen encounter, IgM+ and IgD+ naïve B cells can switch to expression of activated classes IgG, IgA, and IgE via genomic recombination of the immunoglobulin heavy chain constant region locus. Much of current knowledge about the mechanisms of class switching is derived from the analysis of B cells induced to undergo class switch recombination (CSR) in vitro. However, the patterns of antibody class switching in the natural setting within a living organism have remained largely uncharacterized.

How switch recombination is directed to distinct classes in individual cells is a longstanding question (*Esser and Radbruch, 1990*). Cytokine signals, such as CD40 ligand, IL-4, IFNγ, and TGFβ, induce CSR and can direct switching toward specific classes in vitro (*Stavnezer, 1996*). These signals likely originate from cognate Th cells and dendritic cells in vivo. Cytokine stimulation induces transcription and splicing of 'germline' transcripts from the switch region of the particular IGHC locus that is participating in CSR (*Lorenz et al., 1995*; *Stavnezer-Nordgren and Sirlin, 1986*). These switch regions accumulate histone modifications that are associated with open chromatin conformations and high DNA accessibility (*Jeevan-Raj et al., 2011*; *Wang et al., 2009*). Together, these experiments suggest a model in which epigenetic control of switch region accessibility directs CSR toward specific classes (*Alt et al., 1986*; *Stavnezer-Nordgren and Sirlin, 1986*; *Vaidyanathan and Chaudhuri, 2015*).

In this study, we mapped the landscape of human antibody class switching using high-throughput immune repertoire sequencing (*Boyd et al., 2009*; *Weinstein et al., 2009*). This method has previously yielded insights into how the immune system responds to pathogen challenge and vaccination (*Jackson et al., 2014a*; *Jiang et al., 2013*; *Parameswaran et al., 2013*; *Racanelli et al., 2011*; *Vollmers et al., 2013*; *Wang et al., 2014*) and changes with age (*Jiang et al., 2011*, *2013*; *Wang et al., 2014*). We developed an approach for reconstructing clonal histories of antibody

lineages, including class switching events. We used this method to measure antibody class switching within clonal lineages across the entire repertoire in vivo in a cohort of healthy human twins. This comprehensive map identifies how antibodies of every class are created. Our analysis of class switching events within clonal lineages uncovered signatures of the cellular decision processes that direct CSR toward specific isotypes.

## Results

### Antibody repertoire sequencing with subclass resolution

To investigate human antibody class switching in vivo, we conducted immune repertoire sequencing of immunoglobulin heavy chain (IGH) genes of 22 healthy young adult human twins, including 9 pairs of identical twins and 2 pairs of fraternal twins. Sequencing libraries were prepared of a ~430–480 bp fragment of the IGH gene using total RNA from peripheral blood B cells drawn from each subject. Libraries were sequenced with 300 bp paired-end reads on the Illumina Miseq platform. Individual RNA molecules were labeled with unique molecular barcodes during library preparation, enabling highly accurate measurement of genetic diversity by using a consensus read approach to correct PCR and sequencing errors (*Figure 1—figure supplement 1*; *Vollmers et al., 2013*). On average, ~261,000 raw reads were obtained from each individual at each time point, representing ~154,000 unique sequences (*Figure 1—figure supplement 2A*). Molecular barcodes were used to enumerate unique sequences and identify distinct clones. Sequencing reads covered ~100 bp of the constant region, making it possible to determine antibody class and resolve subclasses (IgG1, IgG2, IgG3, IgG4, IgA1, and IgA2) with high accuracy. Across individuals, the most abundant class was IgM (75%), followed by IgG1 (10%) and IgA1 (8%) (*Figure 1—figure supplement 4A* and *Table 1*). For 14 of 22 subjects, the measurement was repeated 28 days after the initial sample as a biological replicate (Bio. Rep.). To test the robustness of these measurements, we used the Jensen-Shannon distance as a measure of similarity between distributions of antibody class abundance. As expected, the class distributions are essentially the same across replicates in nearly every subject (*Figure 1—figure supplement 4A and B*). Furthermore, V gene usage is highly similar across biological replicates (*Figure 1—figure supplement 7*).

### Reconstructing clonal history of antibody lineages

After activation by specific antigen, naïve B cells proliferate and undergo somatic hypermutation and class switching. This process gives rise to a clonally related lineage composed of antibody IGH sequences of distinct classes that also differ in the variable region due to accumulation of somatic mutations. To study class switching after B cell activation, we developed an approach for reconstructing the clonal history of antibody lineages by using the information contained in the accumulation of somatic mutations as a molecular clock, much as ribosomal 16S sequences are used to study the evolutionary relationships of life on earth. We identified sequences belonging to the same clonal

**Table 1.** Number of unique sequences of each class analyzed in this study.

| Class | Sequences (Sample) | Sequences (Bio. Rep.) |
|---|---|---|
| IgM | 2,423,262 | 1,899,952 |
| IgD | 70,169 | 60,510 |
| IgG3 | 16,981 | 15,625 |
| IgG1 | 117,025 | 143,053 |
| IgA1 | 276,189 | 231,477 |
| IgG2 | 213,574 | 176,484 |
| IgG4 | 6,751 | 9,672 |
| IgE | 278 | 262 |
| IgA2 | 63,374 | 62,251 |

lineage as those sharing a variable (V) and joining (J) gene combination, CDR3 length, and ≥95% sequence identity in both the CDR3 and the rest of the variable region with at least one other member of the lineage (*Figure 1—figure supplement 5*, *Figure 2—figure supplement 7*, and Materials and methods). We reconstructed a minimum evolution tree for each clonal lineage by conducting a multiple sequence alignment of all sequences in the lineage and then identifying a minimum spanning tree which includes all the sequences and minimizes the total number of mutations across the tree.

CSR occurs through a genomic rearrangement of the IGH constant region locus that brings the gene segment encoding the new constant region closer to the VDJ locus. Gene segments between the old and new constant regions are looped out and deleted (*Iwasato et al., 1990*; *Schwedler et al., 1990*; *Yoshida et al., 1990*). Therefore, class switches are irreversible and must proceed from upstream classes to downstream classes, according to the order of the IGH constant region loci on the chromosome, which is shown in *Figure 1—figure supplement 3*. This provides a constraint on ancestry that we incorporated into our algorithm for clonal history reconstruction: for a sequence belonging to a given class, only sequences of upstream classes can be ancestors.

The lineage trees are rooted on the germline sequence of the V and J gene combination shared by the lineage. As expected, somatic mutations accumulate as one moves from the germline sequence toward the leaves of the trees (*Figure 1—figure supplement 6A*). We note that PCR and sequencing errors rarely give rise to sequences having different classes and therefore contribute minimally to error in measuring class switching. Because these errors terminate branches of the tree, they also do not affect our analysis of mutation accumulation and correlations in class switching patterns. Unlike previous approaches (*Barak et al., 2008*), our tree reconstruction approach enables direct measurement of class switching events in clonal lineages which are supported by observed sequences without the need to infer mutations or ancestral isotype states. The inference process is challenging and indirect due to the complex mutational spectrum of somatic hypermutation.

Examples of reconstructed clonal histories of activated B cell lineages from one subject are displayed in *Figure 1*. The ~154,000 unique sequences from each subject belonged to ~34,000 distinct clonal lineages on average (*Figure 1—figure supplement 2B*). On average across all subjects, each sequence displays 99.2% identity in the VDJ variable region to its parent (*Figure 1—figure supplement 6B*), suggesting that the repertoire has been sampled deeply enough to enable accurate, high-resolution reconstruction of lineage history, since most pairs of sequences are separated by at most a few sites of hypermutation.

## Measuring the landscape of antibody class switching

To characterize the landscape of antibody class switching in living humans, we measured probabilities of class switching across the entire repertoire. We devised an algorithm that traverses the reconstructed tree for each lineage and counts switches between classes from ancestor to child. We then calculated the relative frequency of switching between every pair of classes. In total ~142,000 class switch events were observed and contributed to this data set.

To characterize the accuracy of this approach, we note that we detected ~35,000 pairs of sequences sharing identical VDJ sequences but having different classes, as indicated by differing constant region sequences. Since every molecule has a unique barcode associated with it, we are able to prove that these sequences are not due to PCR recombination artifacts (*Figure 2—figure supplement 1A*). These sequence pairs arose from CSR without intervening hypermutation events and enable analysis of class switching rates on a subset of the data without the need for lineage identification or tree construction. When we conducted our analysis using only these sequences, we found that the patterns of class switching correlate extremely well with the landscape measured across the entire repertoire, showing that the full lineage tree approach faithfully measures class switching patterns (*Figure 2—figure supplement 1B and C*; *Table 2*). We further confirmed that the patterns of class switching measured using only sequences that inherited all of the germline mutations from their immediate ancestor are highly similar to those measured using the full lineage tree approach (*Figure 2—figure supplement 2*), indicating that artifacts arising from imperfect sampling of ancestral sequences have not distorted our measurement. In addition, patterns of class switching measured using only sequences supported by at least three sequencing reads are highly similar to those measured using the full lineage tree approach, suggesting that PCR and sequencing error have not

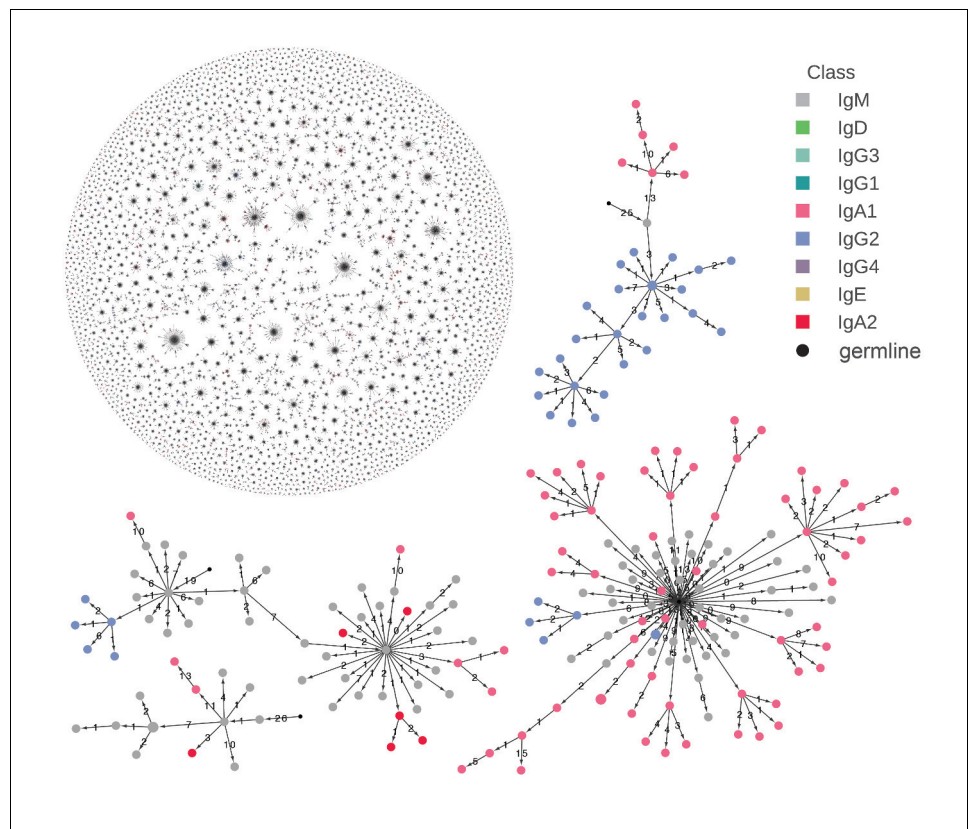

**Figure 1.** Reconstructed clonal histories of B cell lineages. Examples of reconstructed clonal histories of antibody lineages in the repertoire of Subject 1A. All lineages with ≥6 sequences, comprising 64% of unique sequences in the repertoire, are shown in the upper left. Four examples among these lineages are also shown. Circles indicate unique IGH sequences colored by class. Edges indicate the minimum evolution tree that spans the clonal lineage and are labeled with the number of substitutions separating the sequences. The tree is rooted on the germline V and J gene sequence, indicated by the small black circle.

The following figure supplements are available for figure 1:

**Figure supplement 1.** Schematic of immune repertoire sequencing strategy and data processing.

**Figure supplement 2.** Number of unique sequences (**A**) and clonal lineages (**B**) identified in each subject.

**Figure supplement 3.** Schematic of human immunoglobulin heavy chain (IGH) locus.

**Figure supplement 4.** Abundance of antibody classes.

**Figure supplement 5.** Determination of sequence identity cutoff for clonal lineages.

**Figure supplement 6.** Features of reconstructed antibody lineages.

**Figure supplement 7.** V gene usage is similar in biological replicates.

substantially distorted our measurement (*Figure 2—figure supplement 3*). Finally, we confirmed that patterns of class switching could not simply be explained by random switching in proportion to class abundance. After shuffling the classes of parent-child sequence pairs, the hierarchical class switching patterns that we observed vanished (*Figure 2—figure supplement 4*).

To confirm that we measured the antibody repertoire with sufficient depth to accurately characterize the class switching landscape, we performed rarefaction analysis using data from five

**Table 2.** Counts of pairs of sequences sharing identical VDJ sequences, but different constant region sequences. Data from all subjects including both original and biological replicate samples are shown.

| | IgM/IgD | IgG3 | IgG1 | IgA1 | IgG2 | IgG4 | IgE | IgA2 |
|---|---|---|---|---|---|---|---|---|
| IgG3 | 744 | | | | | | | |
| IgG1 | 6440 | 2234 | | | | | | |
| IgA1 | 6374 | 338 | 4676 | | | | | |
| IgG2 | 3210 | 1325 | 5673 | 2291 | | | | |
| IgG4 | 50 | 46 | 129 | 3 | 61 | | | |
| IgE | 2 | 0 | 2 | 4 | 0 | 0 | | |
| IgA2 | 1619 | 30 | 427 | 2581 | 1630 | 2 | 0 | |

subjects. This analysis revealed that the landscape asymptotes rapidly to the observed one as read sampling depth is increased (*Figure 2—figure supplement 5*). Since class switch intermediates that are missing from the data could contribute to error in our measurement, we performed additional rarefaction analysis to show that the fraction of class switches that occur via an intermediate also saturates as read sampling depth is increased (*Figure 2—figure supplement 6*). As further validation, we examined ~1500 sequences that were detected in both biological replicates of the same subject and found that in 99.9% of cases the class of the ancestor sequence was the same in both samples, indicating that the presence or absence of switch intermediates is reliably detected.

## Landscape of antibody class switching

Our measurement of class switching patterns uncovered a hierarchy of pathways leading to the production of antibodies of specific classes, which we have summarized as a state transition diagram showing the relative rates of all possible switches (*Figure 2A*, *Figure 2—figure supplement 8A*, *Table 3*). The dominant class switch pathway leads from IgM/IgD to IgG1 or IgA1. Specifically, IgM switched most commonly to IgG1, IgA1, and IgG2, which together account for ~85% of switches from IgM. Direct switches from IgM to downstream classes (IgG4, IgE, or IgA2) were rare (~14%). Instead, downstream classes are predominantly produced via indirect switches (*Figure 2B*), most often through IgG1 or IgA1 in a secondary hierarchy of pathways. For example, IgG1 frequently switched to IgA1 or IgG2, which together account for ~92% of switches from IgG1, but rarely switched directly to IgG4 or IgA2. Most IgA2 was produced by subsequent switches from IgA1 or IgG2 (~65%), instead of directly from IgG1. We also saw that IgG3 is more likely to switch to IgG1/2 (~87%) rather than IgA1/2, suggesting that IgG3 lies along a pathway for specific generation of IgG antibodies. These results delineate the class switch pathways that give rise to specific antibody classes. The class switching landscape and the penetrance of direct and indirect switches were highly reproducible across the biological replicates, which were separated by 28 days (*Figure 2C*, *Figure 2—figure supplement 8B*, and *Figure 2—figure supplement 9*), confirming the robustness of our measurements and suggesting that the landscape is a temporally invariant feature of the healthy human immune system.

## Variation in class switching landscape between individuals

Next, we examined how the landscape of class switching varies between individuals. The landscapes of individual subjects are broadly similar (*Figure 2—figure supplements 9* and *10*), and the dominant usage of several major switch pathways is conserved across all subjects (*Figure 2—figure supplement 9D*). This similarity is also apparent when measured by the Jensen-Shannon distance, which reveals that the magnitude of variation between subjects is similar to variation between biological replicates of the same subject (*Figure 2—figure supplement 9A and B*). We conclude that healthy young adults share a broadly conserved landscape of antibody class switching.

We also asked whether class switching landscapes were more similar among identical twins compared to unrelated individuals. We found that the class switching patterns of identical twins are no better correlated than pairs of unrelated individuals (*Figure 2—figure supplement 9A–C*),

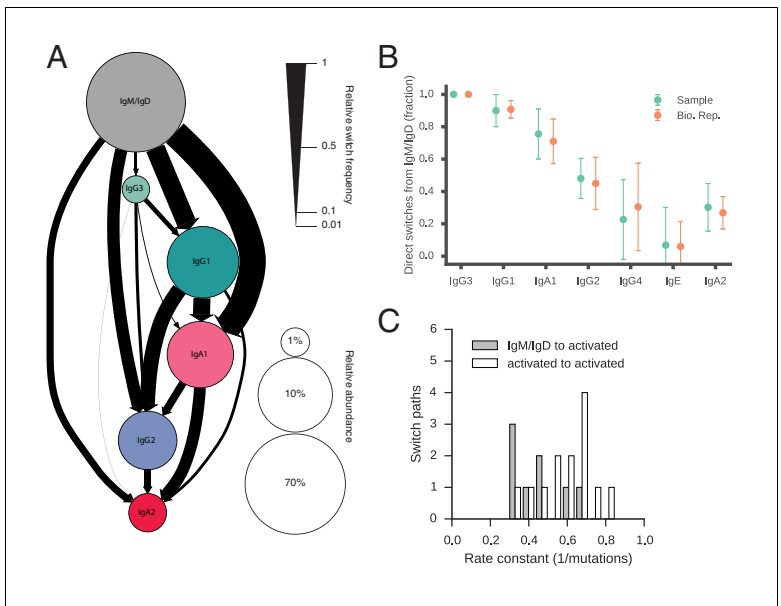

**Figure 2.** Landscape of human antibody class switching. (**A**) State transition diagram of class switching. Classes are indicated as circles and possible switches as arrows. The radius of each circle indicates the relative abundance of the labeled class. The width of each arrow indicates the relative frequency of the switch (also reported in **Table 3**). Rare classes IgG4 and IgE have been omitted for clarity and are shown in **Figure 2—figure supplement 8A**. (**B**) Penetrance of direct switches from IgM/IgD. For each class, the fraction of sequences created by direct switching from IgM is shown (mean ± s.d. across n = 22 subjects for Sample and n = 14 subjects for Bio. Rep.). (**C**) Rates of CSR. The rate constant of each switch path was estimated by fitting an exponential probability distribution to the distribution of the number of somatic mutations accumulated prior to CSR (**Figure 2—figure supplement 11**). Distributions of rate constants for switch paths from IgM/IgD to activated classes (gray) and from an activated class to another activated class (white) having ≥500 examplesin both Sample and Bio. Rep. repertoires are shown.

The following source data and figure supplements are available for figure 2:

**Source data 1.** Counts of class switch events.

**Figure supplement 1.** Patterns of class switching measured using sequences with identical VDJ sequences but different constant regions are highly similar to those measured using the full lineage reconstruction approach.

**Figure supplement 2.** Patterns of class switching measured using sequences inheriting all germline mutations from parent are highly similar to those measured using the full lineage reconstruction approach.

**Figure supplement 3.** Patterns of class switching measured using sequences supported by consensus reads are highly similar to those measured using the full lineage reconstruction approach.

**Figure supplement 4.** Landscape of class switching cannot be explained by random switching in proportion to the abundance of antibody classes.

**Figure supplement 5.** Landscape of class switching saturates with respect to sequencing depth.

**Figure supplement 6.** Rarefaction analysis indicates that switch intermediates are robustly detected.

**Figure supplement 7.** Class switching landscape is not sensitive to the lineage clustering cutoff parameter.

**Figure supplement 8.** Landscape of class switching in humans.

**Figure supplement 9.** Comparisons of class switch landscapes across individuals.

*Figure 2 continued on next page*

*Figure 2 continued*

**Figure supplement 10.** Class switching landscapes of individual subjects.
**Figure supplement 11.** Measurement of rates of class switching.

suggesting that the regulation of CSR involves substantial environmental or stochastic influences, as has been found in many other parameters of the immune system (*Brodin et al., 2015*).

## Class switch recombination from naïve to activated classes is slower than between activated classes

Despite progress in dissecting the molecular mechanisms of CSR, many fundamental characteristics of class switching under physiological conditions remain unresolved. For example, the tempo of class switching within activated B cell lineages has not been measured. Motivated by this, we used somatic mutations as a molecular clock to measure the rate of CSR between naïve and activated classes within clonal lineages. We searched the clonal lineage trees for motifs consisting of multiple sequences sharing the same class that accumulated mutations prior to a class switch event (*Figure 2—figure supplement 11A*). We asked how many somatic mutations accumulate prior to CSR and whether different antibody classes tend to accumulate different numbers of mutations before class switching.

We found that naïve classes (IgM or IgD) accumulated significantly more mutations before undergoing CSR to activated classes (IgG, IgA, or IgE), in comparison with CSR between activated classes. Among IgM/IgD sequences, the average number of mutations accumulated in the variable region prior to CSR is 4.1 ± 6.7 (mean ± s.d.). In contrast, only 2.5 ± 4.3 mutations accumulate in sequences of activated classes prior to further CSR (*Figure 2—figure supplement 11B and C*; p = $1.8 \times 10^{-45}$ and $1.3 \times 10^{-12}$ for Samples and Bio. Rep. respectively; Mann-Whitney U test, two-sided). We found that the distributions of mutations accumulated prior to CSR were well fit by an exponential distribution, suggesting that CSR is a memory-less process with respect to somatic mutation and allowing us to estimate rates of CSR (*Figure 2—figure supplement 11E*). We found that the rate constants of class switching from IgM/IgD to activated classes were ~0.37 mutation$^{-1}$ (*Figure 2C*). By contrast, the rate constants of switching between activated classes were ~0.60 mutation$^{-1}$ (p < $1.5 \times 10^{-4}$; Mann-Whitney U test, two-sided). These patterns were robustly observed

**Table 3.** Number of parent-child pairs having each possible pair of classes. Data from all subjects and both biological replicates are included. Total number of parent-child pairs is 3,304,346. Number in parentheses indicates the relative frequency of each class switch.

| Child | Parent | | | | | | | |
|---|---|---|---|---|---|---|---|---|
| | IgM/IgD | IgG3 | IgG1 | IgA1 | IgG2 | IgG4 | IgE | IgA2 |
| IgM/IgD | 1,495,250 | | | | | | | |
| IgG3 | 2,530 (0.09) | 48,357 | | | | | | |
| IgG1 | 19,881 (0.74) | 4,726 (0.18) | 645,591 | | | | | |
| IgA1 | 26,915 (1.00) | 935 (0.03) | 17,480 (0.65) | 493,999 | | | | |
| IgG2 | 12,695 (0.47) | 3,178 (0.12) | 15,663 (0.58) | 8,028 (0.30) | 375,284 | | | |
| IgG4 | 312 (0.01) | 157 (0.006) | 342 (0.01) | 57 (0.002) | 542 (0.02) | 16,091 | | |
| IgE | 10 (0.0004) | 0 (0) | 16 (0.0006) | 5 (0.0002) | 1 (0.00004) | 3 (0.0001) | 419 | |
| IgA2 | 7,344 (0.27) | 139 (0.005) | 2,934 (0.11) | 12,176 (0.45) | 6,455 (0.24) | 17 (0.0006) | 0 (0) | 86,814 |

across biological replicates (*Figure 2—figure supplement 11D*). Because class switching can be initiated from mutated IgM memory cells for which we may have failed to detect naïve progenitors, our measurements of the number of mutations accumulated in IgM sequences prior to class switching are likely an underestimate, supporting an argument for differences in the rate of CSR between naïve and activated classes. Thus, our results suggest that activated B cells which have already undergone class switching tend to rapidly undergo further class switching, as measured by the clock of somatic hypermutation.

## Class switch fates are strongly correlated among closely related cells and lose coherence as somatic mutations accumulate

The isotype composition of the antibody repertoire is ultimately determined by class switch decisions made by individual B cells, which belong to clonal lineages. However, nothing is known about how class switch fates vary among cells within a clonal lineage. To address this, we traced the descent of individual B cells, using somatic mutations as both lineage markers and a molecular clock, and examined the class switch fates of clonally related cells. We asked whether closely related cells, as measured by the number of somatic mutations accumulated in their sequences since their divergence from a common progenitor, exhibit more concordant class switch fates than more distantly related cells, as well as unrelated cells (from distinct clonal lineages).

To find pairs of related cells, we searched the clonal lineage trees for motifs consisting of a pair of sequences that (1) shared a common progenitor, (2) were the same class, and (3) each had class-switched progeny (*Figure 3—figure supplement 1*). We further required that each sequence inherit all of the somatic mutations present in its ancestor, yielding ~40,000 pairs of sequences for this analysis. We binned these sequence pairs by their mutational distance from the common progenitor, revealing that the pairs spanned a broad spectrum of relatedness (*Figure 3—figure supplement 2*). For each level of relatedness (bin), we calculated the probability that both sequences in a pair switched to the same class. To quantify the strength of concordance, we used Yule's Q, which measures the agreement between pairs of sequences. Yule's Q ranges from -1 to 1, with 1 indicating that both sequences always switched to the same class (perfect agreement), -1 indicating that the two sequences always switched to different classes (perfect disagreement), and 0 indicating no correlation between the fates of sequences in a pair.

We discovered that closely related cells made highly concordant class switch decisions (*Figure 3A*). The most closely related cells, separated by ≤2 mutations from their common progenitor, had a very significant tendency to switch to the same class, in contrast to unrelated pairs of cells which were obtained by shuffling (p values ranging from $6 \times 10^{-5}$ to $1.1 \times 10^{-186}$; *Figure 3—figure supplement 3*). When we examined pairs of cells that were less closely related (as measured by mutations from their common progenitor), we found that the concordance in class switch fates dissipated as somatic mutations accumulated, becoming indistinguishable from unrelated cells after ~10 somatic mutations (*Figure 3B* and *Figure 3—figure supplement 4*). These findings were corroborated by examining the probability distributions of class switch fate conditioned upon the fate of a closely related cell: closely related cells exhibit probability distributions that are strongly biased toward the same class switch fate, and the bias dissipates as somatic mutations accumulate (*Figure 3—figure supplement 5*). Importantly, we found that mutational distances (i.e. branch lengths) are not associated with particular switching events (*Figure 3—figure supplement 6*), and that mutational distances among related sequences are not correlated (*Figure 3—figure supplement 7*), indicating that correlations in switch fate are not due to differential sampling of lineages. These findings demonstrate that class switch decisions are coordinated within clonal lineages of B cells in living humans and that this coordination dissipates at large genealogical distances within a lineage.

## Coordination of class switch fate within clonal lineages is an autonomous property of purified B cells

We reasoned that the coordination of class switch decisions among closely related cells could arise if CSR is directed to specific isotypes by cytokine signals originating from cognate cells in spatially localized niches, which sister cells at some point co-occupy and therefore are directed synchronously toward the same class switch fate. Alternatively, sister cells might share an imprinted state, which is

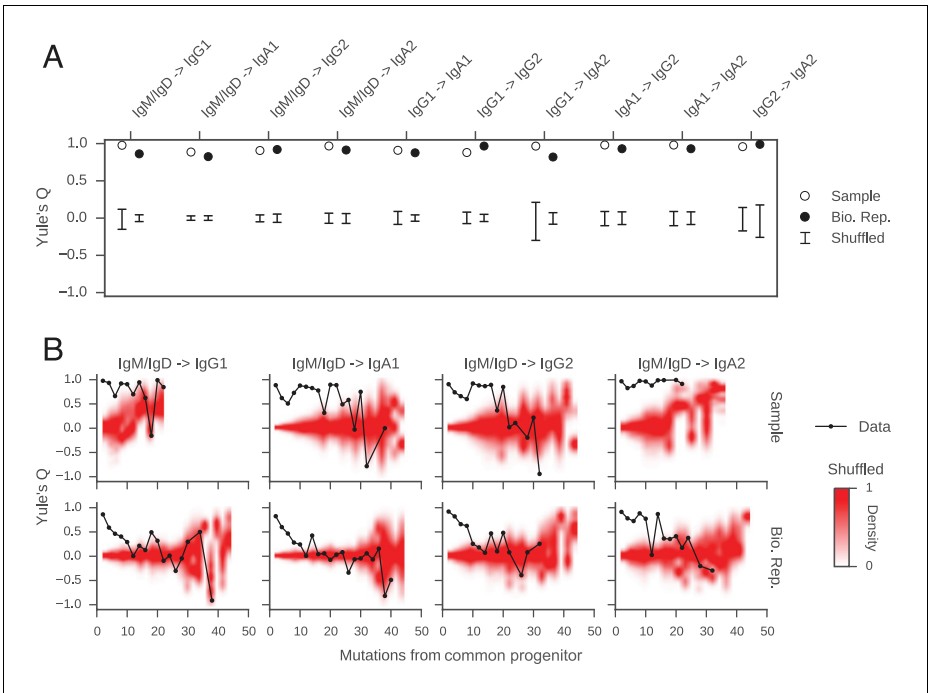

**Figure 3.** Class switch fates of closely related sequences are correlated and lose coherence as somatic mutations accumulate. (**A**) Concordance between the class switch fates of closely related sequences having ≤2 substitutions from their common progenitor, as measured using Yule's Q. Distinct switch paths are indicated on the x-axis. Bars show standard deviation of the concordance Q between pairs of unrelated sequences, which were obtained by shuffling (1000 replicates). (**B**) Concordance between the class switch fates of pairs of sequences plotted by their relatedness, as measured by number of mutations from their common progenitor. For comparison, red shading indicates the probability density of concordance between unrelated sequence pairs obtained by shuffling (1000 replicates). To account for variation due to sampling statistics, the number of pairs at each level of relatedness was preserved during shuffling.

The following figure supplements are available for figure 3:

**Figure supplement 1.** Motif analyzed to characterize the class switch fates of clonally related cells.

**Figure supplement 2.** Relatedness of the pairs of cells used to characterize the class switch fates of clonally related cells in vivo.

**Figure supplement 3.** Estimation of significance of correlations between class switch fates of related sequences.

**Figure supplement 4.** Concordance between the class switch fates of related sequences plotted against relatedness as measured by number of mutations from common progenitor for all common switch paths.

**Figure supplement 5.** Closely related B cells often switch to the same class.

**Figure supplement 6.** Branch lengths are not associated with particular switching events.

**Figure supplement 7.** Mutational distances among related cells, common progenitors, and switched progeny are not correlated.

transmitted from a common progenitor and directs CSR toward specific isotypes in a cell-autonomous fashion. To discriminate between these models and test whether cellular interactions are necessary to generate correlations between class switch fates of sibling cells, we measured the class switch behavior of purified primary human B cells stimulated with cytokines in culture. We purified

CD19+ IgM+ B cells from whole blood (*Figure 4—figure supplement 1A*) and cultured them in the presence of multimeric CD40 ligand (CD40L), IL-4, and IL-10 for 8 days, then prepared sequencing libraries of the IGH locus. The purified B cells were 99.6% CD19+ (*Figure 4—figure supplement 1B*), and sequencing of the IgM+ cells used to initiate the culture revealed that 97% of the sequences were IgM/IgD (*Figure 4A*). During culture, the cells proliferated and underwent class switching to IgG1, IgG2, IgG3, and IgA1 (*Figure 4B*). After reconstructing the histories of clonal lineages, we identified ~1900 pairs of IgM/IgD sequences sharing a common progenitor that subsequently underwent class switching.

In vitro, the concordance between the class switch fates of the most closely related cells was as evident as in vivo (*Figure 4C*). Closely related sequences separated from a common progenitor by ≤2 substitutions exhibited a highly significant tendency to switch to the same class, in comparison with unrelated sequences (p values ranging from $5 \times 10^{-4}$ to $5 \times 10^{-28}$). Since sequences having more somatic mutations were exceedingly rare (*Figure 4—figure supplement 2*), it was not possible to examine the dissipation of concordance as somatic mutations accumulated. Importantly, no pairs of sequences sharing a common progenitor that subsequently underwent class switching were detected in the IgM+ B cell populations that initiated the culture, confirming that imperfect IgM+ B cell purification cannot account for the class switches detected after culturing. These results demonstrate that coordination of class switch decisions in clonal lineages of B cells is not dependent upon the in vivo environment and interactions with cognate cells, but rather seems to be an autonomous property of purified B cells. This finding suggests that CSR is directed toward specific isotypes by an imprinted state, which is transmitted from a common progenitor to sister cells.

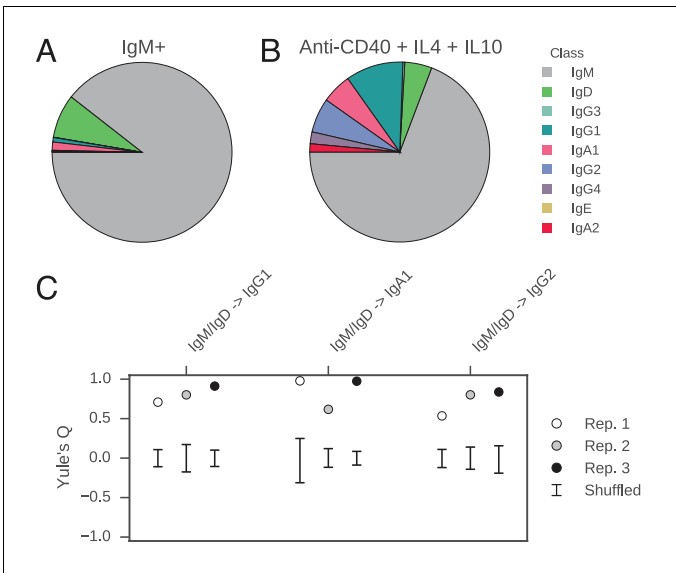

**Figure 4.** Class switch fates of closely related sequences are correlated in purified B cells induced to class switch in vitro. (**A**) Class composition of CD19+ IgM+ cells used to initiate cell culture as measured by sequencing of the IGH locus. (**B**) Class composition of cells after culture for 8 days in the presence of multimeric CD40L, IL-4 and IL-10 (mean of three replicates). (**C**) Concordance between class switch fates of closely related sequences having ≤2 mutations from their common progenitor measured using Yule's Q. Distinct switch paths are indicated on the x-axis. Bars show standard deviation of the concordance Q for unrelated pairs of sequences, which were obtained by shuffling (1000 replicates). Results of three replicate experiments are indicated by color.

The following figure supplements are available for figure 4:

**Figure supplement 1.** Purification of CD19+ IgM+ B cells for in vitro culture.

**Figure supplement 2.** Relatedness of the pairs of cells used to characterize the class switch fates of clonally related cells in vitro.

## Discussion

Deep sequencing of the immune repertoire offers unprecedented views into the human immune system. In this study, we set out to use antibody repertoire sequencing to investigate the nature of antibody class switching in healthy humans. The mechanisms of CSR and each individual's history of immune activation leave indelible imprints on the diversity of immunoglobulin sequences. We have exploited these signatures to characterize patterns of class switching in the natural setting in living humans and dissect the cellular processes that govern CSR.

Our findings provide a comprehensive map of the patterns of antibody class switching in humans. The data reveal that pathways of CSR are organized into two tiers: (1) naïve classes (IgM/IgD) switch predominantly to proximal classes such as IgG3, IgG1, or IgA1, and (2) proximal classes may subsequently switch to distal classes, such as IgG2, IgG4, or IgA2. IgG1 and IgA1 seem to be central intermediates linking naïve to distal classes. This pattern is evident from (1) the high probability of switching from naïve to proximal classes together with the low probability of switching directly from naïve to distal classes, and (2) observations of frequent sequential switches via IgG3, IgG1, and IgA1 intermediates. This hierarchy mirrors the linear geometry of the IGH constant region loci on chromosome 14, suggesting that chromosomal structure or topology influences CSR. Consistent with this, previous studies which showed that mouse B cells stimulated in vitro with CD40L and IL-4 switched to IgG1 with high frequency after three divisions, while switching to the downstream classes IgE and IgG2a increased after five or six divisions (*Hasbold et al., 1998*; *Hodgkin et al., 1996*). A previous study on human populations reported increasing levels of point mutation in progressively further downstream IgG subclasses, supporting sequential class switching (*Jackson et al., 2014b*). This study also suggested the existence of a preferential switch pathway from IgG2 to IgA2 based on evidence of stronger antigen-driven selection in IgA2 than IgA1 sequences, which is consistent with our results. Our examination of clonal histories of B cell lineages with mixed isotypes provides more direct evidence for sequential class switching and the existence of dominant class switch pathways. While our lineage reconstruction approach circumvents the difficulties associated with ancestral inference and probabilistic models of class switching, one limitation is that measurements of switch rates can be affected by undersampling of ancestors, especially for switches between rare classes, such as IgE and IgG4. We have estimated this undersampling rate by using a Chapman estimator to gauge the total number of each isotype in circulation compared to the number we detected, and found results ranging from 0.1% to 5% depending on the subject, which does not affect any of the conclusions of this work.

Prior to this work a limited number of class transitions had been observed in human samples. Our data identifies ten transitions which to our knowledge have not been previously identified (*Table 4*). In conjunction with other published results, it appears that, with the possible exception of IgE, any transition which is permitted to happen by the geometry of the immunoglobulin locus is observed in healthy human samples. This finding supports the view that the machinery underlying CSR is intrinsically stochastic, and that biological regulation enforces probabilistic preferences, rather than strict rules in switch behavior.

The broad contours of the class switching landscape appear to be conserved across individuals, but there is variation between individuals that likely reflects differences in the history of immune activation and environmental exposure. Importantly, identical twins did not exhibit identical class switching landscapes, indicating that class switching is driven largely by non-heritable factors, which likely include exposure to pathogens or other microbes. Previous studies of identical twins have suggested that genetic background controls features of the antibody repertoire, such as IGHV, IGHD, and IGHJ gene use, and CDR3 length (*Wang et al., 2015*). On the other hand, studies examining other components of the immune system have indicated that non-heritable factors dominantly influence most features of serological and cellular responses, including serum protein abundances and cell populations (*Brodin et al., 2015*). Our findings suggest that variation between human in the class composition of the antibody repertoire is predominantly driven by the ability of the immune system to adapt to environmental stimuli, rather than genetic predisposition. Unique landscapes of antibody class switching in identical twins likely arise from the unpredictable stimulation of B cell clones and different exposure to many microbes over the course of a lifetime. Our measurement of the conserved class switching landscape of healthy, young adult humans provides a reference for comparison against individuals with altered immune states, such as autoimmunity or chronic infection.

**Table 4.** Summary of class switch recombination events that have been observed in human cells. Switches that have previously been observed are indicated as 'Known' and the literature references are provided. All of the previous studies demonstrated the existence of switch events by sequencing recombination junctions or switch circles. 'Novel' indicates switches which have not previously been reported that we observed in our dataset of ~35,000 pairs of sequences sharing identical VDJ sequences, but having different constant region genes. 'Not detected' indicates switches that we did not observe in this dataset of identical sequences.

| Destination class | Source class | | | | | | |
| --- | --- | --- | --- | --- | --- | --- | --- |
| | IgM/IgD | IgG3 | IgG1 | IgA1 | IgG2 | IgG4 | IgE |
| IgG3 | Known (*Fujieda et al., 1995*; *Malisan et al., 1996*) | | | | | | |
| IgG1 | Known (*Fujieda et al., 1995*; *Malisan et al., 1996*) | Novel | | | | | |
| IgA1 | Known (*Jabara et al., 1993*; *Zan et al., 1998*) | Known (*Lin et al., 2014*) | Known (*Zan et al., 1998*) | | | | |
| IgG2 | Known (*Malisan et al., 1996*) | Novel | Novel | Novel | | | |
| IgG4 | Known (*Fujieda et al., 1995*; *Jabara et al., 1993*) | Novel | Novel | Novel | Novel | | |
| IgE | Known (*Jabara et al., 1993*; *Xiong et al., 2012*) | Not detected | Known (*Xiong et al., 2012*) | Novel | Not detected | Known (*Jabara et al., 1993*) | |
| IgA2 | Known (*He et al., 2007*; *Lin et al., 2014*) | Known (*Lin et al., 2014*) | Known (*Lin et al., 2014*) | Known (*He et al., 2007*; *Lin et al., 2014*) | Known (*Lin et al., 2014*) | Novel | Not detected |

Our work demonstrates how somatic mutations can be exploited as a molecular clock to reconstruct the genealogies of cells and characterize the dynamics of cell state. We have uncovered strong correlations between the class switch fates of closely related B cells which have undergone maturation in the natural context in living humans. These correlations appear to decay on a timescale of ~10 somatic mutations. Such correlations between closely related cells are also generated during in vitro culture of purified B cell populations in the presence of cytokines that induce class switching, demonstrating that the correlations are an autonomous property of purified B cells and that the in vivo environment is not necessary to create them. Although cytokine signals driving CSR in vivo likely originate from cognate T helper cells and dendritic cells in localized intercellular niches, our experiments show that correlations in class switch fate between sibling cells cannot simply be attributed to exposure to common signals due to co-occupancy of the same niche. This mode of regulation contrasts with stem cell maintenance and differentiation in the mammalian lung, which are regulated by signaling from parent/progenitor cells to daughter cells in localized niches (*Pardo-Saganta et al., 2015*). We note that the correlations that we observed in vitro were not quite as strong as those seen in vivo, suggesting that cellular interactions in the natural context might enhance sibling correlations in class switch fates. We note that in our culture experiments the starting populations were heterogeneous and included both naïve and memory B cells, leaving the lineage characteristics of class switching within these compartments to be examined in future work.

Our data suggest a model where CSR is directed toward specific classes by a transient epigenetic state, which is transmitted from parent cells to daughter cells and relaxes on a timescale of ~10 somatic mutations. Consistent with this, directed CSR is thought to be regulated via cytokine-activated transcription at specific IGHC loci, which targets the region for modification by activation-induced cytidine deaminase (AID). Germline transcription of IGHC genes is associated with histone modifications that increase DNA accessibility (*Jeevan-Raj et al., 2011*; *Wang et al., 2009*), and germline IGHC transcripts can form RNA-DNA hybrids with genomic DNA, exposing ssDNA to AID attack (*Reaban and Griffin, 1990*; *Reaban et al., 1994*; *Yu et al., 2003*). Using single-cell transcriptomics, we have found that single B cells stimulated to class switch in vitro often predominantly express germline transcripts from a single IGHC locus (F. Horns, unpublished data). We propose that inheritance during mitosis of germline transcripts and chromatin state in the IGHC locus, including

perhaps histone modifications influencing DNA accessibility, is a mechanism that generates correlations in the class switch fates of sister cells. Importantly, epigenetic inheritance of active and repressed chromatin state during mitosis has been demonstrated (*Cavalli and Paro, 1998*; *Grewal and Klar, 1996*). Our measurements suggest that the timescale on which the relative accessibility of IGHC loci persists is ~10 somatic mutations. Calibration of the mutational clock should allow recovery of information about epigenetic state and phenotypic dynamics in units of time and cellular generations. Together with recent studies of mammalian (*Spencer et al., 2009*) and bacterial cells in culture (*Hormoz et al., 2015*), our work suggests that phenotypic correlations between sister cells due to shared inheritance are widespread. We predict that such correlations often will be detected when genealogical relationships between individual cells can be resolved. We propose that inheritance of epigenetic state provides a mechanism for orchestrating cellular behavior without the need for signaling.

## Materials and methods

### Twin cohort

All study participants gave informed consent and protocols were approved by the Stanford Institutional Review Board. Twenty-two human twins aged 18–28, including 11 males and 11 females, were recruited in 2010. All subjects were apparently healthy and showed no signs of disease. Twin zygosity was determined by short tandem repeat analysis with 18 loci. Monozygosity was assigned when all loci and the gender-determining marker were identical.

### Sample collection, PBMC isolation, and RNA extraction

Blood was drawn from each subject by venipuncture. Peripheral-blood mononuclear cells (PBMCs) were isolated using a Ficoll gradient and frozen in 10% (vol/vol) DMSO/40% (vol/vol) fetal bovine serum (FBS) following Stanford Human Immune Monitoring Center protocols. After cells were thawed, total RNA was extracted using the Qiagen AllPrep kit (Valencia, CA). Subjects were vaccinated with the 2010 seasonal trivalent inactivated influenza vaccine immediately after the sample was drawn. Biological replicates were drawn 28 days later. Biological replicates were indistinguishable from the original samples with respect to antibody class and V gene usage (*Figure 1—figure supplements 4B* and *7*), as expected given that the most pronounced immune response occurs 7 days after vaccination (*Wrammert et al., 2008*).

### Library preparation

Sequencing libraries were prepared using 500 ng of total RNA as input following the protocol described in (*Vollmers et al., 2013*). Briefly, primer annealing to a pooled set of ten isotype-specific

**Table 5.** IGH constant region primers.

| Name | Primer (5' to 3') |
| --- | --- |
| IgA_08N | TGACTGGAGTTCAGACGTGTGCTCTTCCGATCTNNNNNNNNGGGGAAGAAGCCCTGGAC |
| IgA_12N | TGACTGGAGTTCAGACGTGTGCTCTTCCGATCTNNNNNNNNNNNNGGGGAAGAAGCCCTGGAC |
| IgG_08N | TGACTGGAGTTCAGACGTGTGCTCTTCCGATCTNNNNNNNNGGGAAGTAGTCCTTGACCA |
| IgG_12N | TGACTGGAGTTCAGACGTGTGCTCTTCCGATCTNNNNNNNNNNNNGGGAAGTAGTCCTTGACCA |
| IgM_long_8N | TGACTGGAGTTCAGACGTGTGCTCTTCCGATCTNNNNNNNNGAAGGAAGTCCTGTGCGAG |
| IgM_long_12N | TGACTGGAGTTCAGACGTGTGCTCTTCCGATCTNNNNNNNNNNNNGAAGGAAGTCCTGTGCGAG |
| IgE_long_8N | TGACTGGAGTTCAGACGTGTGCTCTTCCGATCTNNNNNNNNAAGTAGCCCGTGGCCAGG |
| IgE_long_12N | TGACTGGAGTTCAGACGTGTGCTCTTCCGATCTNNNNNNNNNNNNAAGTAGCCCGTGGCCAGG |
| IgD_long_8N | TGACTGGAGTTCAGACGTGTGCTCTTCCGATCTNNNNNNNNTGGGTGGTACCCAGTTATCAA |
| IgD_long_12N | TGACTGGAGTTCAGACGTGTGCTCTTCCGATCTNNNNNNNNNNNNTGGGTGGTACCCAGTTATCAA |

IGH constant region primers that contain 8 or 12 random nts (*Table 5*) was carried out at 72°C for 3 min then immediately placed on ice for 2 min. First-strand cDNA synthesis was performed using Superscript III reverse transcriptase (Life Technologies, Carlsbad, CA) according to manufacturer's instructions. Second-strand cDNA synthesis was done using Phusion HiFi DNA polymerase (Thermo Scientific, Waltham, MA) and a pool of six IGH variable region primers that contain 8 random nts (*Table 6*) (98°C for 4 min, 52°C for 1 min, 72°C for 5 min). Double-stranded cDNA was purified twice using Ampure XP beads (Beckman Coulter, Indianapolis, IN) at a 1:1 ratio, then amplified using Platinum HiFi enzyme (Life Technologies) and primers containing Illumina adapters and dual indexes. PCR products were purified once using Ampure XP beads at a 1:1 ratio then pooled for multiplexed sequencing.

## Sequencing and data preprocessing

High-throughput sequencing was performed on the Illumina MiSeq platform (Illumina, San Diego, CA) with 300 bp paired end reads. Reads were passed through a pipeline to construct consensus sequences from reads containing the same 16 nt random barcode similar to (*Vollmers et al., 2013*). Base quality scores in the consensus were calculated from the error probabilities associated with bases in the raw reads. Sequences were annotated with V and J germline gene usage and CDR3 length using IgBlast (*Ye et al., 2013*). Classes were determined using BLASTN against a custom database of IGH constant region fragments. We have deposited the sequencing reads in the Sequence Read Archive (accession number PRJNA324281). Preprocessed sequence data and custom analysis scripts are available for download (doi:10.5061/dryad.bv989).

## Lineage clustering

Sequences belonging to the same clonal B cell lineage were identified using clustering as follows. Sequences sharing the same V-J combination and CDR3 length were grouped. Within each group, clusters were found by performing single linkage clustering with a cutoff of 95% sequence identity across both the CDR3 and the rest of the variable region. Sequence identity was computed from ungapped pairwise alignments by counting mismatches. Stringent quality filtering was implemented by assuming mismatches at positions at which the base in either aligned sequence had $Q \leq 5$.

To choose the optimal cutoff for identifying sequences in clonal lineages, we examined the distributions of pairwise sequence identity within groups of CDR3 sequences sharing the same V and J gene combination and CDR3 length (*Figure 1—figure supplement 5*). By plotting the identity of each sequence to the most similar sequence in its group (its 'nearest neighbor'), we saw that sequences separate into two groups: sequences with a highly similar nearest neighbor (>90% identity) and sequences that are substantially dissimilar to the nearest neighbor (40–80% identity). This pattern suggests that the first group consists of sequences that belong to a clonal lineage, while the second group consists of singleton sequences. Thus using a stringent cutoff of 95% sequence identity in the CDR3 ensures that the identified lineages contain sequences that are clonally related. In order to show that the results are not sensitive to the particular choice of a cutoff value we repeated subsequent analyses with cutoffs varying from 80% to 95% and obtained nearly identical results (*Figure 2—figure supplement 7*).

**Table 6.** IGH variable region primers.

| Name | Primer (5' to 3') |
| --- | --- |
| Primer1_1_70 | ACACTCTTTCCCTACACGACGCTCTTCCGATCTNNNNNNNNSCAGCTGGTGCAGTCTGG |
| Primer1/3/5_70 | ACACTCTTTCCCTACACGACGCTCTTCCGATCTNNNNNNNNGTGCAGCTGGTGGAGTCTG |
| Primer2 | ACACTCTTTCCCTACACGACGCTCTTCCGATCTNNNNNNNNTCACCTTGAAGGAGTCTGG |
| Primer4_1 | ACACTCTTTCCCTACACGACGCTCTTCCGATCTNNNNNNNNTGCAGCTGCAGGAGTCG |
| Primer4_2 | ACACTCTTTCCCTACACGACGCTCTTCCGATCTNNNNNNNNGTGCAGCTACAGCAGTGG |
| Primer6 | ACACTCTTTCCCTACACGACGCTCTTCCGATCTNNNNNNNNGTACAGCTGCAGCAGTCA |

## Rarefaction analysis

Rarefaction of preprocessed sequences, which represent molecules of IGH mRNA, was performed by selecting a fraction of sequences uniformly at random from the sequences prior to lineage clustering using a custom script written in Python.

## Reconstructing clonal histories of lineages

The clonal history of each lineage was reconstructed using a custom algorithm. An ungapped multiple alignment of sequences in each lineage was performed by aligning the anchor sequences that mark the start and end of the CDR3. The concatenated sequences of the V and J germline genes were then added to this alignment by performing a profile-profile alignment using MUSCLE with options '-maxiters 2 –diags' which introduces a gap corresponding to the D gene and untemplated nucleotides. A pairwise distance matrix was constructed by counting the number of substitutions required to transform each aligned sequence into the others. This matrix defines a weighted graph of possible ancestor-child relationships. A constraint on ancestry based on antibody class was then applied by pruning edges that violate the geometry of the IGH constant region locus. The minimum evolution tree was identified by using Edmonds' algorithm to find a minimum spanning tree on this directed graph (*Edmonds, 1967*). The tree was rooted on the germline sequence.

## Measuring class switching landscapes

Class switching events were identified by traversing the minimum evolution tree for each lineage and searching for ancestor-child pairs of sequences having different classes. The probabilities that define the class switching landscape were calculated as follows. Relative switch frequency is equal to the number of switches observed from A to B divided by the total number of switches. Destination probability, which describes the probability that a cell switching from the ancestral class will choose the downstream class as the destination, is equal to the number of class switches from class A to class B divided by the total number of switches exiting class A. Similarly, arrival probability, which describes the probability that a cell of a given downstream class originated from the ancestral class, is equal to the number of class switches from class A to class B by the total number of switches entering class B. To generate shuffled ancestor-child pairs, we performed sampling without replacement on the list of ancestor classes to assign an ancestor class to each child class.

## Calculating twin correlations

To quantify correlations between twins, we developed a generalization of the intraclass correlation coefficient (*Shrout and Fleiss, 1979*) for multidimensional data. We treat each measurement as an n-dimensional vector, where n is the number of probabilities measured per twin. To calculate the between- and within-target variance, we computed the distance from the mean vector using the squared Euclidean norm. Intraclass correlation is then equal to $(B - W)/(B + W)$, where B is the between-target variance and W is the within-target variance. To calculate intraclass correlation for twins, the twin pairs are treated as the within-target pairs. For unrelated individuals, all possible pairs of unrelated individuals are treated as the within-target pairs.

## Analysis of rates of class switching

To quantify the amount of hypermutation between class switching events, we searched in the clonal lineage trees for motifs consisting of multiple parent-child pairs sharing the same class prior to a class switch event. We filtered for motifs in which each child sequence inherited all of the somatic mutations relative to the V and J germline genes that were present in its parent, yielding ~74,000 switches for this analysis. The number of mutations accumulated before CSR was calculated by summing the number of substitutions separating parent-child pairs prior to the class switching event. Fitting of an exponential distribution to the empirical distribution of the number of mutations prior to class switching was performed using Python and SciPy (*Oliphant, 2007*) in the IPython environment (*Perez and Granger, 2007*).

## Analysis of correlations in class switch fates among related cells

To explore the inheritance of class switching fates, we searched in the trees for motifs in which two clonally related sequences sharing the same class (1) descended from a common progenitor of the

same class and (2) subsequently switched to different classes. We further filtered for motifs where each descendant sequence inherited all of the somatic mutations relative to the V and J germline genes present in its ancestor. We binned these sequence pairs by relatedness measured by the number of substitutions separating the sequences from the common progenitor, using the maximum number among the two sequences. We then examined the downstream class to which each sequence switched. To quantify concordance, we calculated Yule's Q from the odds ratio describing whether both sequences switched to the same downstream class. Focusing on a single downstream class, let $a$ be the number of cases where both sequence 1 and sequence 2 switched to this class, $d$ be the number of cases where both sequence 1 and sequence 2 did not switch to this class, and $b$ and $c$ be the number of cases where sequence 1 switched to this class, but sequence 2 did not, and vice versa, respectively. Then the odds ratio OR is $(ad)/(bc)$ and Yule's Q is $(OR - 1) / (OR + 1)$. We also examined the conditional probabilities describing the class switch fate of one sequence given the class switch fate of the other sequence.

## Cell culture

We obtained whole blood drawn from volunteers at the Stanford Blood Center and prepared enriched B cell fractions using the RosetteSep kit (StemCell Technologies, Cambridge, MA) according to manufacturer's instructions. We sorted CD19+ IgM+ cells and cultured them at $5 \times 10^5$ cells/ml for 5 days at 37 C and 5% $CO_2$ in RPMI 1640 with L-glutamine (ThermoFisher) supplemented with 10% fetal bovine serum, 10 mM HEPES pH 7.4, 0.1 mM non-essential amino acid (Sigma-Aldrich, St. Louis, MO), 1 mM sodium pyruvate, 100 μ/ml penicillin, 100 μg/ml streptomycin (ThermoFisher), 40 μg/ml apo-transferrin, 500 ng/μl multimeric CD40 ligand (Miltenyi Biotec, San Diego, CA), 200 ng/ml IL-4 (Sigma-Aldrich), and 200 ng/ml IL-10 (Sigma-Aldrich). We extracted RNA from the cells using the RNeasy Micro Kit (Qiagen) according to manufacturer's instructions, but omitting the DNase digestion step. We then prepared sequencing libraries using 24.5 ng of total RNA as input as described above, except that PCR products were purified using Ampure XP beads at a 0.65:1 ratio instead of a 1:1 ratio before pooling for multiplexed sequencing. We processed the sequences, reconstructed the clonal lineage histories, and measured correlations between the class switch fates of related cells as described above.

## Acknowledgements

We thank our study volunteers for their participation in this study. Thanks to SLVP vaccine study staff for conducting the clinical study: research nurses Sue Swope and Tony Trela; CRAs Ashima Goel, Sushil Batra, Isaac Chang, Kyrsten Spann, Raquel Fleischmann; and phlebotomist Michele Ugur. We also thank Lolita Penland for help with cell culture experiments; Christopher J Emig for discussions; and Norma Neff, Gary Mantalas and Ben Passarelli (Stanford Stem Cell Genome Center) for assistance with sequencing and computational infrastructure. This research was supported by the National Science Foundation Graduate Research Fellowship (to FH) and NIH U19A1057229 (to MMD). This work was also supported in part by the Clinical and Translational Science Award UL1 RR025744 for the Stanford Center for Clinical and Translational Education and Research (Spectrum) from the National Center for Research Resources, National Institutes of Health.

## Additional information

### Funding

| Funder | Grant reference number | Author |
|---|---|---|
| National Science Foundation | Graduate Research Fellowship | Felix Horns |
| National Institutes of Health | U19A1057229 | Mark M Davis |

The funders had no role in study design, data collection and interpretation, or the decision to submit the work for publication.

## Author contributions
FH, Designed the study, Performed cell culture experiments, Developed pipeline for sequence analysis, Analyzed data, Wrote the manuscript; CV, Prepared sequencing libraries from human samples; DC, Developed pipeline for sequence analysis; SFM, GES, CLD, Coordinated subject recruitment and sample collection; MMD, Designed the study; SRQ, Designed the study, Analyzed data, Wrote the manuscript

## Author ORCIDs
Felix Horns, http://orcid.org/0000-0001-5872-5061
Derek Croote, http://orcid.org/0000-0003-4907-1865

## Ethics
Human subjects: All study participants gave informed consent and protocols were approved by the Stanford Institutional Review Board.

# Additional files

## Major datasets
The following datasets were generated:

| Author(s) | Year | Dataset title | Dataset URL | Database, license, and accessibility information |
|---|---|---|---|---|
| Felix Horns | 2016 | Data from: Lineage Tracing of Human B Cells Reveals the In Vivo Landscape of Human Antibody Class Switching | http://dx.doi.org/10.5061/dryad.bv989 | Available at Dryad Digital Repository under a CC0 Public Domain Dedication |
| Felix Horns | 2016 | Immunoglobulin heavy chain sequencing | http://www.ncbi.nlm.nih.gov/bioproject/PRJNA324281/ | Publicly available at NCBI BioProject (accession no: PRJNA324281) |

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
