## [Decision Letter]

Thank you for submitting your article "Lineage Tracing of Human B Cells Reveals the in vivo Landscape of Human Antibody Class Switching" for consideration by *eLife*. Your article has been reviewed by three peer reviewers, and the evaluation has been overseen by a Reviewing Editor and Arup Chakraborty as the Senior Editor. The reviewers have opted to remain anonymous.

The reviewers have discussed the reviews with one another and the Reviewing Editor has drafted this decision to help you prepare a revised submission.

Summary:

The manuscript uses sequencing reads of the variable region and parts of the surrounding constant region of immunoglobulin heavy chains to investigate the dynamics of class switch recombination (CSR) in humans. The data are used to infer the relative rates of all possible class switch events. The authors further demonstrate that class switch recombination is correlated within clonal B cell lineages and that this correlation decreases with increasing distance of the variable part of the sequence. All reviewers agreed that these are interesting results addressing an important question. However, we identified various points that require further discussion or additional analysis.

Essential revisions:

1) It is not clear how errors were corrected using the molecular barcodes. You state that you obtained ~261k reads per sample which typically represented 170k distinct molecules (subsection “Antibody Repertoire Sequencing with Subclass Resolution”, first paragraph). This would suggest that most barcodes are represented only once and no consensus sequence of (>=3 sequences) can be computed. Did you use singleton and doubleton barcodes as Figure 1—figure supplement 1 suggests? If so, sequencing errors likely contribute to diversity in many of the clonal clusters (the inserts are quite long and the end of the 2nd read often has low sequencing quality, hence sequencing errors are a concern).

While it seems plausible that errors rarely produce sequences that have different classes, sequencing errors can produce additional sequences that are more similar to a sequence of a different class and hence inflate CSR rate estimates. This should be clarified and the analysis should be carefully controlled for the influence of sequencing errors. Ideally, only proper consensus variants should be used.

2) The authors have opted for a custom analysis pipeline using minimal spanning trees instead of phylogenetic reconstruction and probabilistic models of class switching. The analysis doesn't account for the possibility of unsampled ancestors and it is not clear how the assumption that all ancestors are sampled influences the inference of CSR events. The problem of unsampled ancestors is well known when reconstructing transmission histories of viruses (e.g. http://dx.doi.org/10.1371/journal.pcbi.1003397). The authors performed a number of analysis to validate the CSR inference (presented in Figure 2—figure supplement 1: restriction to identical VDJs, rarefaction analysis), but some concerns remain. A combination of a Z -> X and a Z -> Y transition with an unsampled Z will be misinterpreted as an X -> Y transition, regardless of whether X and Y are separated by somatic mutations or not. For transitions with a small number of counts, a few such instances could contribute substantially. The rarefaction analysis is not informative for transitions with small rates and few counts.

The authors should either provide additional analyses that address these concerns, or state the limitations of their rate inference explicitly.

3) You claim that class switching landscapes of identical twins are not identical, but are they significantly more different than those inferred from biological replicates? If not, the section heading "Identical Twins Do Not Share Identical Class Switching Landscapes" is misleading.

4) The analysis of concordant class switch events on subtrees of two parents with a common ancestor in the same state which go on to produce progeny of different classes is elegant, but we would like to see an additional control: Is the distance between children to parents correlated with that of parents to the common progenitor? Such correlations could arise due to differential sampling of lineages. In this case, similarities in switching might be due to similar distances of parents to children. In other words: Are the length of branches associated with particular switching events? And if so, are branch length of sisters correlated? This can be readily checked.

We would also like you to clarify whether you can conclude that there is a cell specific state that favors specific CSR events, or whether the observed correlation between closely related cells can be explained by an overall variation of in the switching rate, which will mostly result in switches to the same state. An analysis of the conditional probabilities of switching, given the switch of the sister (as mentioned in the subsection “Analysis of Correlations in Class Switch Fates among Related Cells”) could address this. More detail on how Yule's Q is calculated should be given and its meaning briefly explained in the main text.

5) Analysis scripts, code, and data need to be made publicly available. Ideally, documented scripts (the authors used python, iPython notebooks might be an option) along with preprocessed data should be provided. In particular, in the light of possible improvements of the analysis, preprocessed data sets that facilitate analysis by others would be welcome. Raw data needs to be deposited in short read archives.

6) The caption of Figure 2—figure supplement 1 refers to Figure 2, but Figure 2 doesn't contain similar matrices. The caption of Figure 2—figure supplement 2 refers to pies and colored slices, but all pies are mono-chrome. *eLife* has no limit on the number of figures. We suggest reorganizing the supplementary figures into smaller more coherent units. The current multi-panel figures with very small print and long captions are not helpful.

---

## [Author Response]

Essential revisions:

*1) It is not clear how errors were corrected using the molecular barcodes. You state that you obtained ~261k reads per sample which typically represented 170k distinct molecules (subsection “Antibody Repertoire Sequencing with Subclass Resolution”, first paragraph). This would suggest that most barcodes are represented only once and no consensus sequence of (>=3 sequences) can be computed. Did you use singleton and doubleton barcodes as Figure 1—figure supplement 1 suggests? If so, sequencing errors likely contribute to diversity in many of the clonal clusters (the inserts are quite long and the end of the 2nd read often has low sequencing quality, hence sequencing errors are a concern).*

While it seems plausible that errors rarely produce sequences that have different classes, sequencing errors can produce additional sequences that are more similar to a sequence of a different class and hence inflate CSR rate estimates. This should be clarified and the analysis should be carefully controlled for the influence of sequencing errors. Ideally, only proper consensus variants should be used.

When the analysis is performed using only sequences supported by >= 3 reads, for which a consensus-read approach could be used to correct PCR and sequencing errors, we found that the patterns of class switching measured are highly similar to those measured using all sequences, including singleton and doubleton barcodes (R^2 ranging from 0.86 to 0.98). This suggests that PCR and sequencing error have not substantially distorted our measurements. We have added a supplemental figure showing these results (Figure 2—figure supplement 3) and noted these results in the main text in the subsection “Measuring the Landscape of Antibody Class Switching”.

*2) The authors have opted for a custom analysis pipeline using minimal spanning trees instead of phylogenetic reconstruction and probabilistic models of class switching. The analysis doesn't account for the possibility of unsampled ancestors and it is not clear how the assumption that all ancestors are sampled influences the inference of CSR events. The problem of unsampled ancestors is well known when reconstructing transmission histories of viruses (e.g. http://dx.doi.org/10.1371/journal.pcbi.1003397). The authors performed a number of analysis to validate the CSR inference (presented in Figure 2—figure supplement 1: restriction to identical VDJs, rarefaction analysis), but some concerns remain. A combination of a Z -> X and a Z -> Y transition with an unsampled Z will be misinterpreted as an X -> Y transition, regardless of whether X and Y are separated by somatic mutations or not. For transitions with a small number of counts, a few such instances could contribute substantially. The rarefaction analysis is not informative for transitions with small rates and few counts.*

The authors should either provide additional analyses that address these concerns, or state the limitations of their rate inference explicitly.

We agree that properly accounting for unsampled ancestors is an important issue and we appreciate the opportunity to clarify this point. In the example raised, independent somatic mutations are likely to have accumulated along the Z ->X and Z -> Y branches. Therefore, one can exclude most cases where the X -> Y transition is incorrectly inferred due to failure to sample Z by examining only sequences where all somatic mutations present in X are inherited by Y. We repeated our analysis of class switch landscapes using only pairs of sequences which satisfy this condition. We called somatic mutations relative to the germline V and J genes to ensure that the ancestral state is known with high confidence. We found that the patterns of class switching measured using only these sequences were highly similar to those measured using all sequence pairs from the full lineage tree approach (R^2 ranging from 0.96 to 1). This suggests that artifacts arising from imperfect sampling of ancestral sequences have not substantially distorted our measurements. We have added a supplemental figure showing these results (Figure 2—figure supplement 2) and noted these results in the main text in the subsection “Measuring the Landscape of Antibody Class Switching”.

Nevertheless, we acknowledge that undersampling of ancestors gives rise to limitations in our rate inference. We performed an additional analysis to estimate the extent of undersampling by estimating the number of each isotype in circulation compared to the number we detected. We found that 0.1 to 5% of each isotype in circulation was sampled, which does not affect our conclusions. We added an explicit discussion of this limitation in the main text in the Discussion.

3) You claim that class switching landscapes of identical twins are not identical, but are they significantly more different than those inferred from biological replicates? If not, the section heading "Identical Twins Do Not Share Identical Class Switching Landscapes" is misleading.

There is a trend toward larger differences between twins compared with biological replicates, but this was not statistically significant (P = 0.59 and P = 0.24 for relative switch frequency and destination probability respectively). Because the variation between twins is similar to the variation between biological replicates separated by 28 days, we agree with the reviewers that the claim that “identical twins do not share identical class switching landscapes” is misleading. Indeed, an individual’s class switching landscape is not identical with his or her own landscape measured 28 days later. Nevertheless, our data show that class switching patterns of identical twins are no better correlated than pairs of unrelated individuals. We have modified the language in the main text in the subsection “Variation in Class Switching Landscape between Individuals” to more clearly state this claim.

*4) The analysis of concordant class switch events on subtrees of two parents with a common ancestor in the same state which go on to produce progeny of different classes is elegant, but we would like to see an additional control: Is the distance between children to parents correlated with that of parents to the common progenitor? Such correlations could arise due to differential sampling of lineages. In this case, similarities in switching might be due to similar distances of parents to children. In other words: Are the length of branches associated with particular switching events? And if so, are branch length of sisters correlated? This can be readily checked.*

We would also like you to clarify whether you can conclude that there is a cell specific state that favors specific CSR events, or whether the observed correlation between closely related cells can be explained by an overall variation of in the switching rate, which will mostly result in switches to the same state. An analysis of the conditional probabilities of switching, given the switch of the sister (as mentioned in the subsection “Analysis of Correlations in Class Switch Fates among Related Cells”) could address this. More detail on how Yule's Q is calculated should be given and its meaning briefly explained in the main text.

We agree with the reviewers that the possibility that correlations in class switch fate arise due to differential sampling of lineages should be carefully tested. As suggested, we have checked whether mutational distances (i.e. branch lengths) are associated with particular switching events, and found that they are not. We also checked whether mutational distances among related sequences are correlated, and found that they are not (R^2 ranging from 0 to 0.19). These results show that correlations in switch fate are not due to differential sampling of lineages. We have added supplemental figures showing these results (Figure 3—figure supplement 6 and Figure 3—figure supplement 7) and noted these findings in the main text in the section “Class Switch Fates are Strongly Correlated among Closely Related Cells and Lose Coherence as Somatic Mutations Accumulate”.

We believe that there is a cell state that favors specific CSR events. To show this more clearly, we have plotted the conditional probability of switch fates given the switch fate of sister, as suggested (Figure 3—figure supplement 5). These plots show that the probability distribution of switch fate is strongly biased toward the class to which the sister switches, and the bias dissipates as somatic mutations accumulate. This change in the probability distribution is consistent with the existence of a cell state that favors a particular class switch outcome. On the other hand, we have found no evidence for an overall variation in switch rate that is correlated between clonally related cells, as indicated by the lack of correlations between branch lengths of related cells (Figure 3—figure supplement 7). We have added a supplemental figure (Figure 3—figure supplement 5) and clarified this point in the main text in the section “Class Switch Fates are Strongly Correlated among Closely Related Cells and Lose Coherence as Somatic Mutations Accumulate”.

As suggested, we have expanded the explanation of Yule’s Q when it is first introduced to the reader in the main text. We have also included additional detail about how to calculate Yule’s Q in the Materials and methods section.

5) Analysis scripts, code, and data need to be made publicly available. Ideally, documented scripts (the authors used python, iPython notebooks might be an option) along with preprocessed data should be provided. In particular, in the light of possible improvements of the analysis, preprocessed data sets that facilitate analysis by others would be welcome. Raw data needs to be deposited in short read archives.

We have uploaded our preprocessed data consisting of annotated immunoglobulin heavy chain (IGHC) sequences and their clonal lineage histories to a web server where they can be downloaded freely. Several example scripts accompany these data and additional scripts are available upon request. We have also deposited the raw sequencing reads in the Sequence Read Archive. We have provided links to these resources in the Materials and methods section.

*6) The caption of Figure 2—figure supplement 1 refers to Figure 2, but Figure 2 doesn't contain similar matrices. The caption of Figure 2—figure supplement 2 refers to pies and colored slices, but all pies are mono-chrome. eLife has no limit on the number of figures. We suggest reorganizing the supplementary figures into smaller more coherent units. The current multi-panel figures with very small print and long captions are not helpful.*

As suggested, we have reorganized all of the supplementary figures to be smaller, coherent units that will be clearer for readers. We have fixed the captions of both figures mentioned.